# Folic Acid Fortification and Neural Tube Defect Risk: Analysis of the Food Fortification Initiative Dataset

**DOI:** 10.3390/nu12010247

**Published:** 2020-01-18

**Authors:** Michaela E. Murphy, Cara J. Westmark

**Affiliations:** 1Nutritional Sciences, University of Wisconsin-Madison, Madison, WI 53706, USA; memurphy6@wisc.edu; 2Department of Neurology, University of Wisconsin-Madison, Madison, WI 53706, USA

**Keywords:** folate, Food Fortification Initiative, methylenetetrahydrofolate reductase (MTHFR), national fortification, neural tube defect, vitamin B9, vitamin B12

## Abstract

The United States implemented mandatory fortification of cereal grains with folic acid in 1998 to prevent neural tube defects (NTDs) during pregnancy. The health benefits of folate (vitamin B9) are well documented; however, there are potential risks of exceeding the upper tolerable limit, particularly in vulnerable populations. We conducted a population-based analysis of the Food Fortification Initiative dataset to determine the strength of the evidence regarding reports of decreased NTDs at the national level in response to mandatory folic acid fortification of cereal grains. We found a very weak correlation between NTD prevalence and the level of folic acid fortification, irrespective of the cereal grain fortified (wheat, maize or rice). Stratification of the data based on socioeconomic status (SES) indicated a strong linear relationship between reduced NTDs and better SES. We conclude that national fortification with folic acid is not associated with a significant decrease in the prevalence of NTDs at the population level.

## 1. Introduction

Neural tube defects (NTDs) are a heterogeneous group of structural birth defects that arise from a complex array of genetic and environmental factors and adversely affect the structure and function of the brain and spinal cord [1]. The United States of America (USA) was the first country to mandate a national folic acid food fortification program to prevent NTDs including spina bifida. Currently, more than 80 other countries fortify cereal grains with folic acid. Based on rates from the Centers for Disease Control and Prevention (CDC) Birth Defects Monitoring Program from 1980 through 1987, an estimated 13,600 infants born in the USA had spina bifida without anencephaly [2]. Of these, approximately 3800 died as a result of their defects [2].

Periconceptional intake of folic acid reduces a women’s risk of having an infant affected by an NTD [3]. Folic acid intervention studies in pregnant women with prior NTD-affected pregnancies showed a 60–100% reduction in NTD risk with a later pregnancy. Observation studies of folic acid efficacy in preventing NTD in pregnant women without a prior NTD-affected pregnancy showed a 0–75% reduction in risk [4]. Thus, in 1992, the USA Public Health Service (PHS) recommended that all women of childbearing age consume 400 μg of folic acid daily to prevent NTDs [4]. Specifically, they stated, “All women of childbearing age in the United States who are capable of becoming pregnant should consume 0.4 mg of folic acid per day for the purpose of reducing their risk of having a pregnancy affected with spina bifida or other NTDs. Because the effects of higher intakes are not well known but include complicating the diagnosis of vitamin B12 deficiency, care should be taken to keep total folate consumption at <1 mg per day, except under the supervision of a physician.” [4].

Despite the publicized guidelines, there was still concern that women of childbearing age were not consuming enough folate and were at risk of NTD-affected pregnancies. To promote compliance, the USA Food and Drug Administration (FDA, Silver Spring, MD, USA) issued a regulation in 1996 requiring that all enriched cereal grain products be fortified with folic acid by January of 1998 [5,6]. Cereal grains were required to be fortified with folic acid at levels ranging from 0.43 mg to 1.4 mg per pound. At the time, the federal agencies recognized that high folate intake was masking vitamin B12 deficiency-associated anemia, which leads to neurological damage if untreated. There was also uncertainty regarding the lack of data on the minimal level of folate needed to significantly reduce NTDs. Despite these concerns, the PHS found the evidence sufficiently consistent to make its recommendation because 50% of pregnancies are unplanned and the neural tube closes early in embryonic development (gestation day 28); thus, folic acid fortification should commence prior to becoming pregnant. They concluded that “the intakes that are likely to result from the level of fortification established in their final rule do not present a health concern to the general population”, and it was estimated that there would be a 50% reduction in NTDs in the USA.

The efficacy of national folic acid fortification programs in reducing NTDs is debated, and there has been an increase in the prevalence of individuals exceeding the tolerable upper limit (UL) for folic acid intake (1000 μg/day) [7,8]. The UL for folic acid was designated as one-fifth of the lowest observed dose associated with a potential adverse outcome (5000 μg/day). The aim of this study was to retrospectively address the question of whether folic acid fortification improves NTDs at the population level by extracting and analyzing relevant data from the Food Fortification Initiative (FFI, Atlanta, GA, USA) dataset.

## 2. Materials and Methods

We utilized the FFI database found online at ffinetwork.org to extract data on folic acid fortification levels and the prevalence of NTDs per 10,000 births as a function of country. The FFI is a global partnership composed of public, private, and civic members that was founded in 2002 to promote fortification of industrially milled flours (wheat, maize, and rice). Members include the CDC’s National Center for Chronic Disease Prevention and Health Promotion, Emory University, and the International Federation for Spina Bifida Hydrocephalus. Funding partners include the Bill & Melinda Gates Foundation (Seattle, WA, USA), Australian Department of Foreign Affairs and Trade (Barton, Australia), Global Alliance for Improved Nutrition (Geneva, Switzerland), Micronutrient Initiative (Ottawa, Ontario, Canada), United Nations International Children’s Emergency Fund (UNICEF, New York, NY, USA), Cargill, Inc. (Minneapolis, MN, USA), GiveWell (Oakland, CA, USA), The Amit J. & Vicky L. Patel Foundation (Atlanta, GA, USA), and Nutrition International (Ottawa, Ontario, Canada). Under the website link for country profiles, the FFI database lists which nutrients including folic acid are added to grains via fortification in units of parts per million (ppm) based on data from the Food and Agriculture Organization of the United Nations (FAO, Roma, Italy) using 2013 data, which was the last year with all data available in March 2018. The FFI attained NTD data from a combination of sources [9,10,11,12,13]. Thus, the FFI dataset contains a single value for folic acid fortification levels for each country and a single value for NTD prevalence.

There was a total of 236 countries monitored by the FFI. Of those countries, data on NTDs were available for 194 countries. Inclusion data consisted of all countries with available data on both folic acid fortification levels and NTD prevalence (186 countries). Eight countries were excluded that were listed as fortifying with folic acid at 0 ppm (Antigua and Barbuda, Bahamas, Barbados, Dominica, Guyana, Philippines, Suriname, and Tajikistan). The study compared countries with national folic acid fortification versus countries without national folic acid fortification with the rationale that a national fortification policy would directly affect the vast majority of inhabitants of a particular country despite the import of grain products from other countries. The null hypothesis was that there is no association between national folic acid fortification and the prevalence of NTDs. The alternative hypothesis was that national folic acid fortification alters the prevalence of NTDs. The primary endpoint of interest examined was prevalence of NTDs. The primary predictor variable was folic acid fortification. To calculate the average prevalence of NTDs per 10,000 births as a function of folic acid fortification, countries were binned into groups based on which cereal grain was fortified and NTD prevalence was summed and divided by the number of countries for each fortification group. Potential confounders included variations in consumption of folic acid, genetic variants, and overall nutritional status at the individual subject level, the accuracy of data reporting agencies, voluntary fortification with folic acid, periconception folic acid supplementation, and variable implementation timing of mandatory folic acid programs. These confounders could not be addressed with the available data. Study size was dependent on the available data in the FFI database.

For analysis of socioeconomic status, UNICEF economic indicator data were merged with the FFI dataset. The percentage of the gross domestic product (GDP) spent on health, education, and social protection were summed, ranked, and divided into quintiles. The UNICEF government expenditure data were based on the most recent year available between 2010–2018. Five countries (Cook Islands, North Korea, Libya, Niue, Syrian Arab Republic) were excluded from the socioeconomic status (SES) analysis because there were no data available on health, education or social protection government expenditures.

Data were analyzed in accordance with STROBE guidelines (https://strobe-statement.org/index.php?id = available-checklists). Means, standard deviations, regression coefficients, and 95% confidence intervals (CI) were computed to describe the results. To statistically test for differences in NTD rates as a function of folic acid fortification, the Student’s t-test was used. Statistical significance was defined as *p* < 0.05.

## 3. Results

### 3.1. Description of Data Utilized from the Food Fortification Initiative (FFI)

There are discrepant reports regarding the efficacy of national folic acid fortification. We conducted a retrospective analysis of the FFI dataset. The FFI tracks food fortification levels for 236 countries. Of these countries, data on both the prevalence of NTDs and the level of folic acid fortification were available for 186 countries (Table 1). There are two countries that fortify wheat, maize, and rice (United States and Costa Rica). There are 14 countries that fortify wheat and maize but not rice, and six countries that fortify wheat and rice but not maize. All the countries that fortify maize also fortify wheat, except for Rwanda, and all of the countries that fortify rice also fortify wheat, except for Bangladesh.

### 3.2. Prevalence of NTDs as a Function of Folic Acid Fortification in the FFI Dataset

The average prevalence of NTDs per 10,000 births in countries that do not fortify any cereal grains with folic acid was 13.32 (SD: 5.50, *n* = 116 countries), and the average prevalence of NTDs in countries with at least one cereal grain fortified with folic acid was 13.30 (SD: 6.13, *n* = 70). Stratification of the data based on which cereal grain was fortified with folic acid indicated no statistically significant differences in the prevalence of NTDs comparing wheat, maize, and rice fortification versus no fortification, albeit there was a trend for reduced NTDs with folic acid fortification of maize (*p* = 0.065) (Table 2, Figure 1, Figure 2, Figure 3 and Figure 4). There was a very weak correlation between NTD prevalence and the level of folic acid fortification irrespective of the cereal grain fortified: wheat (Figure 1), maize (Figure 2), and rice (Figure 3).

There was a trend for about 20% reduced NTDs when maize was fortified with folic acid compared to no fortification (*p* = 0.065) although the level of fortification did not correlate with the prevalence of NTDs. Seventeen countries fortified maize with folic acid at a range of 0.5–2.6 ppm, with the United States at 1.87 ppm. The lower prevalence of NTDs with maize fortification was due to the absence of greater than 15 NTDs per 10,000 births in all 17 countries in the maize cohort. The top quintile for prevalence of NTDs in the wheat and rice cohorts was >15 per 10,000 births with the highest reported prevalence of 32 per 10,000 births for both cohorts.

It should be noted that India and Guatemala had the lowest reported fortification levels of wheat and rice at 0.1 ppm and 0.4 ppm folic acid, respectively, which are more than two standard deviations lower than the mean. If India and Guatemala are deleted from the analysis, then the regression coefficients are −0.16 and 2.06, respectively.

### 3.3. Prevalence of NTDs after Stratification of the Data Based on Socioeconomic Status

An indicator of SES is the percent of GDP spent on healthcare, education, and social protection. Stratification of the FFI NTD data based on these national economic indicators produces a strong linear correlation of reduced NTDs with higher SES (a = 1.43; R^2^ = 0.85). The average prevalence of NTDs in quintile 1 (highest SES) was 10.97 (4.83) versus quintile 5 at 16.11 (6.47) (*p* = 0.0003; greater than 30% reduced prevalence). Binning the countries by folic acid fortification status gives a near perfect linear relationship between improved SES and reduced prevalence of NTDs when cereal grains are fortified with folic acid (a = 2.05; R^2^ = 0.996), and a moderate correlation between SES and NTDs with the non-fortification cohort (a = 1.19; R^2^ = 0.55) (Table 3). Interestingly, the quintile 3 countries had a higher average prevalence of NTDs with folic acid fortification (*p* < 0.03). In total, these data strongly suggest that improved SES contributes to reduced prevalence of NTDs.

## 4. Discussion

Mandatory folic acid fortification programs in the USA, Canada, Costa Rica, Chile, and South Africa are associated with significant increases in blood folate concentrations and declines of 25%–50% in the prevalence of NTD-affected pregnancies [14,15,16,17,18,19]. Reported NTDs in the USA decreased from 10.8/10,000 births in 1995–1996 to 6.9/10,000 births in 2006 [14]. A systematic review (179 studies) and meta-analysis (123 studies) covering the prevalence of spina bifida in response to folic acid fortification status, geographic region, and study population indicate a lower prevalence of spina bifida in geographic regions with mandatory folic acid fortification (33.86 per 100,000 live births) versus voluntary fortification (48.35 per 100,000 live births) [20,21]. Based on 59 countries meeting the criteria of mandatory folic acid fortification of at least 1.0 ppm, it is estimated that 50,270 spina bifida and anencephaly births were prevented out of a possible 280,500 [22]. Overall, these reports suggest that national folic acid fortification protects against NTDs. However, two important confounding issues with these studies are that they do not take into account that NTD rates were declining prior to folic acid fortification and do not include comparison to non-fortification control groups during the same time period.

The annual numbers of NTD-births in the USA and the United Kingdom declined without mandatory fortification [2,23,24]. In the USA, the CDC analyzed data from 16 states for the prevalence of spina bifida at birth between the years 1983–1990 and found 4.6 cases of spina bifida per 10,000 births. The peak of 5.9 cases per 10,000 births in 1984 declined to 3.2 cases per 10,000 births in 1990. Rates varied substantially by state and racial/ethnic groups with the lowest prevalence for Asians/Pacific Islanders and the highest prevalence for Hispanics although the rate for Hispanics declined substantially from 1983–1990. Thus, in the absence of mandatory folic acid fortification, the prevalence of NTDs was substantially decreasing in the USA. It is not possible to discern if that trend would have continued in the absence of national mandatory folic acid fortification.

Health Canada mandated national folic acid fortification by early 1998 consistent with the FDA deadline, which allowed the export of Canadian flour to the USA [25]. Retrospective cross-sectional studies indicate increased folic acid status in both young and older women and an approximately 50% reduction in NTDs. There were no federal or provincially designated studies implemented to prospectively monitor blood folate levels or the prevalence of NTDs. Over one million obstetrical deliveries of all liveborn or stillborn infants were studied in a retrospective population-based longitudinal study between April 1990 and March 2000 to assess whether the Canadian folic acid fortification program was associated with a decline in the rate of pre-eclampsia (PET) and all hypertensive disorders of pregnancy [26]. No significant decline in the monthly rate of either PET or all hypertensive disorders of pregnancy was found after fortification, albeit here was a slight increase in the risk of all hypertensive disorders of pregnancy [26]. While a decline in open NTDs was observed after fortification, the prevalence of NTDs was declining before fortification [26].

To address the confounding issue of a lack of a control group in observational folic acid fortification studies, we analyzed the whole FFI dataset at the population level comparing countries that fortified cereal grains with folic acid to countries not fortifying. We did not find an association between national folic acid fortification and decreased prevalence of NTDs, 13.32 (5.50) per 10,000 births (no fortification) versus 13.30 (6.13) per 10,000 births (plus fortification). There are many reasons why NTDs could decrease over time irrespective of folic acid fortification; for example, improved health care or socioeconomic conditions. We found a strong linear relationship between reduced NTDs and increased GDP spent on socioeconomic indicators (a = 1.43; R^2^ = 0.85). Our findings suggest that improved NTD outcomes are not associated with mandatory folic acid fortification at the population level but rather with SES as indicated by >30% reduced prevalence of NTDs between the lowest and highest SES quintiles. Thus, increased GDP spent on healthcare and education would be expected to improve NTDs. It remains to be determined if improved NTD outcomes as a function of SES are due to periconception folic acid supplementation.

A Cochrane systematic review assessed the efficacy of folic acid fortification of wheat and maize flour on health outcomes in the overall population [27]. The review included 10 studies (five randomized control clinical trials (RCTs), three non-RCTs, and two interrupted time series (ITS) studies) conducted in upper-middle-income countries (China, Mexico, South Africa), a lower-middle-income country (Bangladesh), and a high-income country (Canada). The duration of the intervention studies ranged from 2 weeks to 36 months. The ITS studies included post-fortification periods for up to seven years. The authors concluded that, “Fortification of wheat flour with folic acid may reduce the risk of neural tube defects; however, this outcome was only reported in one non-RCT. Fortification of wheat or maize flour with folic acid (i.e., alone or with other micronutrients) may increase erythrocyte and serum/plasma folate concentrations. Evidence is limited for the effects of folic acid-fortified wheat or maize flour on haemoglobin levels or anaemia. The effects of folic acid fortification of wheat or maize flour on other primary outcomes assessed in this review is not known. No studies reported on the occurrence of adverse effects. Limitations of this review were the small number of studies and participants, limitations in study design, and low-certainty of evidence due to how included studies were designed and reported.” The Cochrane Database of Systematic Reviews (CDSR) is the leading journal and database for systematic reviews in health care, and a contributing author on the review was from the FFI, which tracks global progress in grain fortification. Thus, evidence at the population level is “very low certainty” regarding the efficacy of folic acid fortification in improving NTD outcomes.

While the USA and Canada mandated folic acid fortification in 1998, the United Kingdom, Ireland, and the European Union did not because of the uncertain risk of folic acid fortification to ageing populations who are at increased risk of vitamin B12 deficiency as well as the ethical issue of mandating food fortification when there is a possible risk that fortification might cause risk to a group of the population different from that receiving the benefits [24,28]. In the absence of prospective monitoring of fortification programs, it is not possible to establish a cause and effect relationship on the risks and benefits of national folic acid fortification. Legislation and monitoring guidelines on food fortification for 72 countries as of 31 January 2015 indicate that there is sufficient documentation in terms of establishing mandatory programs but a lack of documentation concerning product compliance to national standards [29]. Thus, surveillance systems are in place to monitor national fortification policies, but not actual exposure and downstream biological effects [30]. Mandatory folic acid fortification in the USA was projected to increase the average folic acid intake by 100 μg/day; however, the mean increase was approximately twice as large as projected [6]. The prevalence of individuals that exceed the UL for folic acid intake is 10% in the subset of the USA population that consumes folic acid supplements [5].

There are several vulnerable populations that may be affected from national folic acid fortification. There is a general risk of nervous system damage in persons with low or insufficient status of B12 who are exposed to high intake of folic acid [7,24]. Low vitamin B12 levels in the elderly, in conjunction with higher serum folate, are associated with anemia and cognitive impairment [5,31,32,33,34,35]. Elderly persons are more likely to take medications such as proton pump inhibitors (PPIs) for heartburn, which are associated with vitamin B12 deficiency, and elderly persons with a generic variant in the transcobalamin gene TCN2 (C776G) who consume twice the recommended daily allowance (RDA) of folate are seven times more likely to have neuropathy [36]. In addition to masking vitamin B12 deficiency, other potential consequences of mandatory folic acid fortification include exceeding the UL of folic acid consumption and folic acid interaction with genetic mutations such as the methylenetetrahydrofolate reductase (MTHFR) polymorphism.

The limitations of this study include: (1) it is based on retrospective data regarding folic acid fortification and the prevalence of NTDs; (2) the level of folic acid fortification is declarative; (3) there are only single data points available for folic acid fortification levels and NTD prevalence (year 2013); and (4) there is a lack of data regarding confounding issues such as voluntary fortification with folic acid, national recommendations for preconception and first trimester folic acid intake, and the implementation timing of mandatory folic acid programs. Despite these limitations, we would expect the main result of the study to be similar for the years before and after 2013. However, without prospective monitoring of folic acid intake and the prevalence of NTDs while taking into account confounding issue such as SES and genetic variations, one cannot definitely prove the findings.

## 5. Conclusions

In conclusion, we did not find reduced prevalence of NTDs at the population level in response to national folic acid fortification, but rather stratification of the data based on SES indicated a strong linear relationship between reduced NTDs and better SES.

## Figures and Tables

**Figure 1 nutrients-12-00247-f001:**
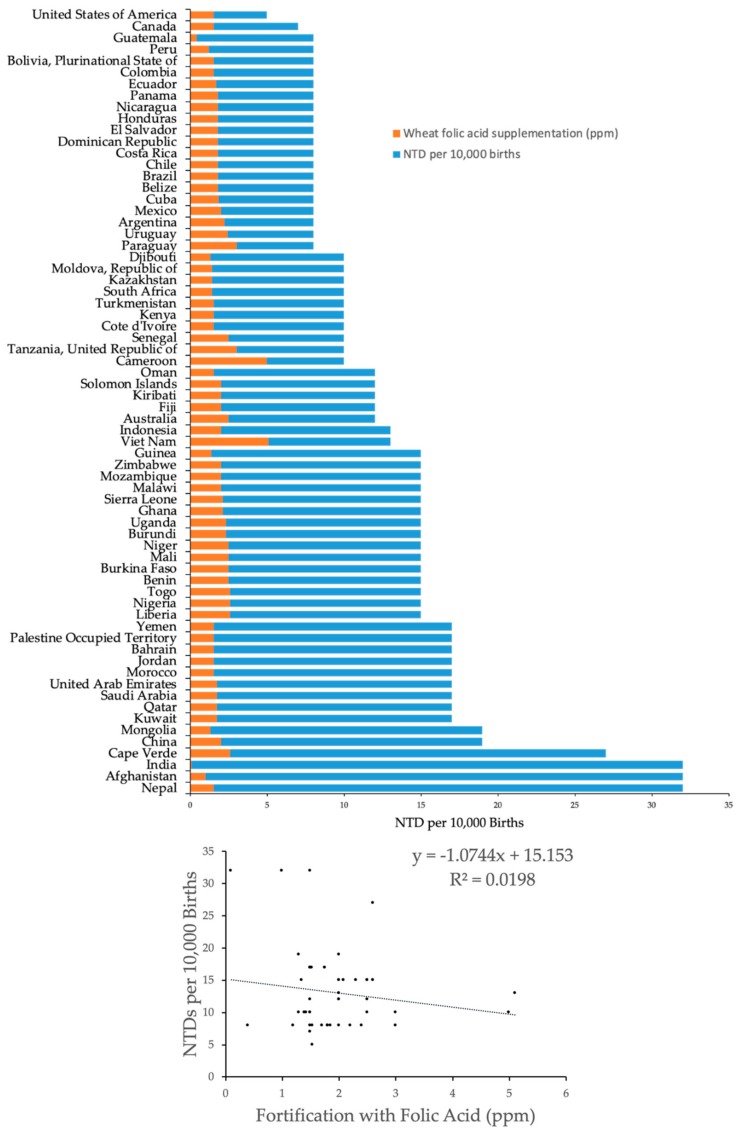
Prevalence of NTDs as a function of folic acid fortification levels in wheat. The number of NTDs per 10,000 births was plotted (blue bars) versus country (*n* = 68). Folic acid fortification levels of wheat in ppm (orange bars) were superimposed on NTD prevalence. Linear regression analysis indicates a regression coefficient (a) of −1.07.

**Figure 2 nutrients-12-00247-f002:**
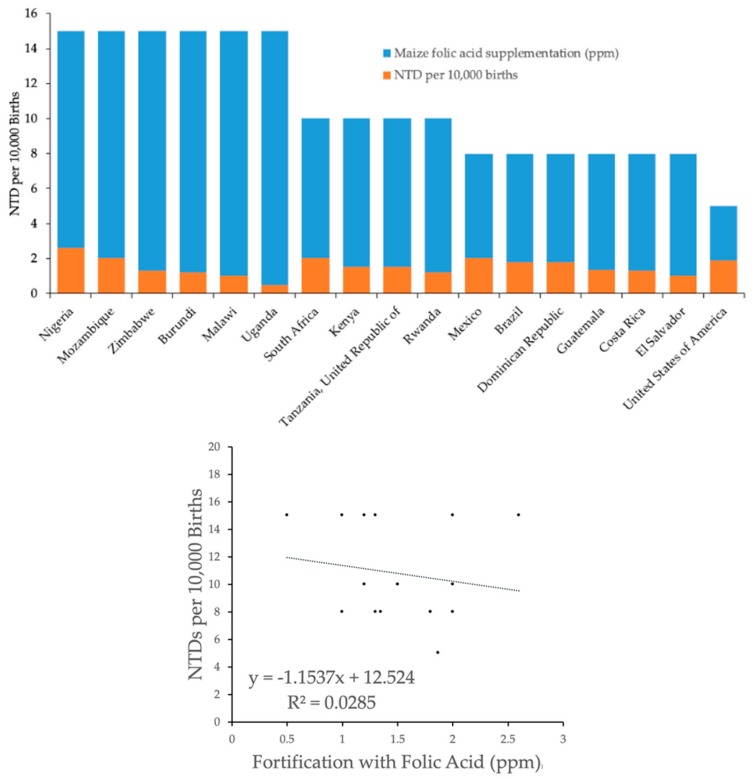
Prevalence of NTDs as a function of folic acid fortification levels in maize. The number of NTDs per 10,000 births was plotted (blue bars) versus country (*n* = 17). Folic acid fortification levels of maize in ppm (orange bars) were superimposed on NTD prevalence. Linear regression analysis indicates a regression coefficient (a) of −1.15.

**Figure 3 nutrients-12-00247-f003:**
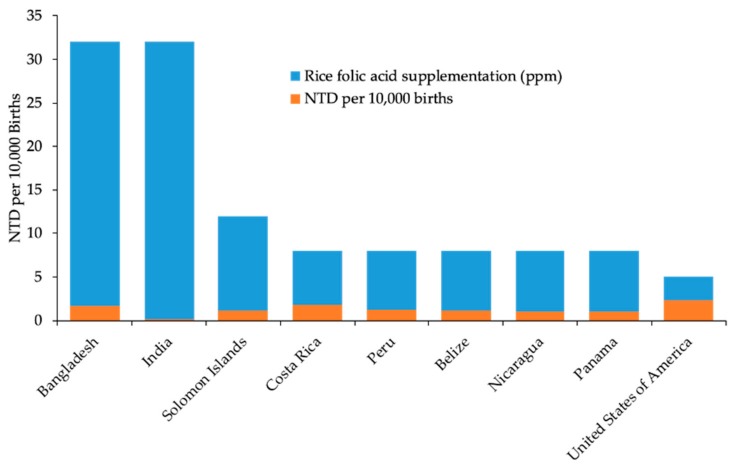
Prevalence of NTDs as a function of folic acid fortification levels in rice. The number of NTDs per 10,000 births was plotted (blue bars) versus country (*n* = 17). Folic acid fortification levels of rice in ppm (orange bars) were superimposed on NTD prevalence. Linear regression analysis indicates a regression coefficient (a) of −6.57.

**Figure 4 nutrients-12-00247-f004:**
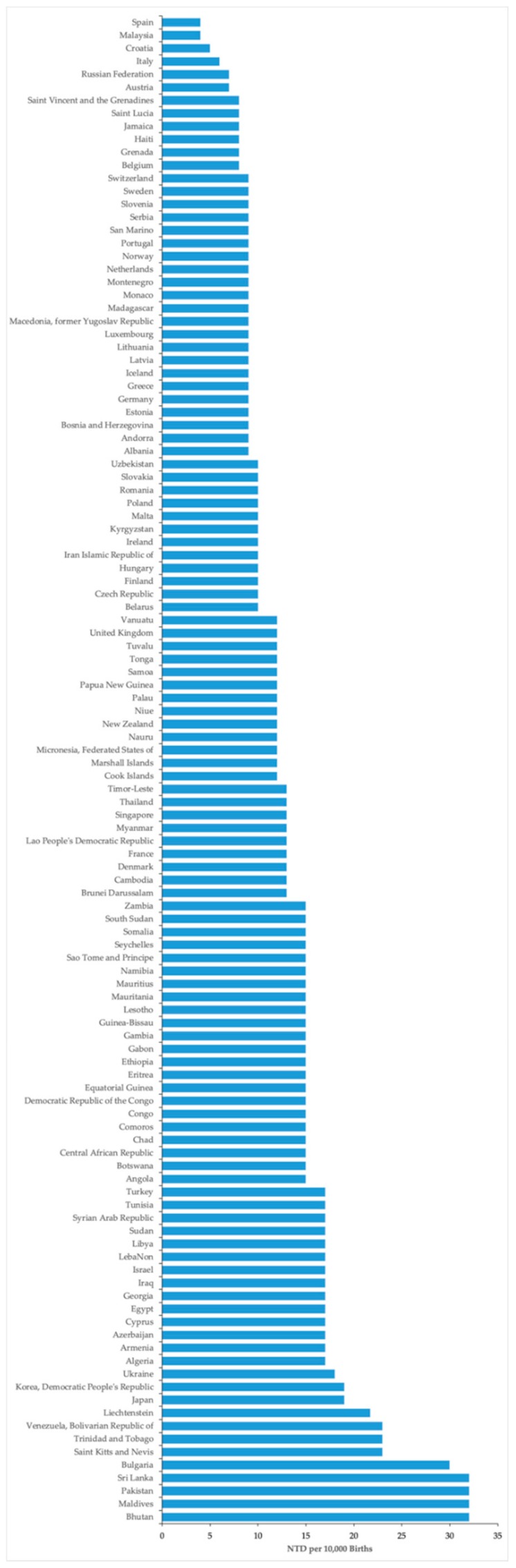
Prevalence of NTDs in the absence of national folic acid fortification. The number of NTDs per 10,000 births was plotted (blue bars) versus country (*n* = 116).

**Table 1 nutrients-12-00247-t001:** Fortification levels of folic acid as a function of cereal grain.

Cereal Grain Fortified	Number of Countries	Average Fortification Level (ppm) ± Standard Deviation ^1^	Range of Fortification (ppm) ^2^
Wheat	68	1.94 (0.75)	0.1–5.11
Maize	17	1.52 (0.50)	0.5–2.6
Rice	9	1.25 (0.63)	0.1–2.31
None	116	*n*/a	*n*/a

^1^ Data extracted from the Food Fortification Initiative dataset at www.ffinetwork.org. Average fortification levels in parts per million (ppm) were calculated by summing country folic acid fortification levels as a function of the cereal grain fortified and dividing by the number of countries in that cohort. Standard deviation is presented in parentheses following the average. ^2^ The range of fortification is the low and high values for each cereal grain cohort in ppm.

**Table 2 nutrients-12-00247-t002:** Prevalence of neural tube defects (NTDs) as a function of folic acid fortification.

Cereal Grain Fortified	*n*	Average Number of NTDs per 10,000 Births ^1^	*p* ^2^	Regression Coefficient ^3^	95% CI ^4^
Wheat	68	13.07 (5.76)	0.78	−1.07	−1.44–1.94
Maize	17	10.76 (3.44)	0.065	−1.15	−0.16–5.28
Rice	9	13.44 (10.67)	0.95	−6.57	−4.21–3.97
None	116	13.32 (5.50)	*n*/a	*n*/a	*n*/a

^1^ Data extracted from the Food Fortification Initiative dataset at www.ffinetwork.org. Average number of NTDs per 10,000 births was calculated by summing NTD prevalence as a function of fortified cereal grain and dividing by the number of countries in that cohort. Standard deviation is presented in parentheses following the average. ^2^ The *p*-value is the Student’s t-test result comparing the indicated cereal grain with the non-fortified cohort. ^3^ The regression coefficient is the constant (a) from the regression line y = ax + b that represents the rate of change of NTDs (y) as a function of folic acid fortification (x). ^4^ CI is the 95% confidence interval for the difference in means compared with the non-fortified cohort.

**Table 3 nutrients-12-00247-t003:** Prevalence of NTDs as a function of socioeconomic status (SES) and folic acid fortification.

SES Quintile ^1^	N	Average NTDs With Fortification ^2,3^	N	Average NTDs Without Fortification ^2,3^	*p* ^4^
1	10	8.90 (2.81)	27	11.74 (5.22)	0.11
2	15	10.87 (5.15)	22	12.45 (5.64)	0.39
3	19	13.11 (3.33)	18	10.50 (3.59)	0.03
4	16	15.38 (7.14)	21	16.19 (5.69)	0.70
5	9	16.90 (6.74)	24	15.82 (6.49)	0.68

^1^ Data extracted from United Nations International Children’s Emergency Fund (UNICEF) dataset at https://data.unicef.org/resources/dataset/sowc-2019-statistical-tables/. ^2^ Data extracted from the Food Fortification Initiative dataset at www.ffinetwork.org. ^3^ Average number of NTDs per 10,000 births was calculated by summing NTD prevalence of countries, as a function of SES quintile, and dividing by the number of countries in that quintile. Standard deviation is presented in parentheses following the average. ^4^ The *p*-value is the Student’s t-test result comparing the fortified and non-fortified countries within the quintile.

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
