# Peer review of "Folic Acid Fortification and Neural Tube Defect Risk: Analysis of the Food Fortification Initiative Dataset"

_nutrients, 2020, doi:10.3390/nu12010247_

Round 1

Reviewer 1 Report

The authors address a global and important issue and are applauded for lots of work which has been put into the preparation of this manuscript. The introduction is very nicely written, but the methods and results could have been better conducted. The discussion is too long and addresses many topics not related to the aims and results of this work. The results of the study have been poorly interpreted in the discussion. Many contents included in the discussion would serve well for a separate review.

Some of the major issues with the methodology include:

Not specifying the timeframe for data collection. The year of introduction of folic acid supplementation varies between countries and the reader is not informed if NTD prevalence refers to recent data or data collected over many years, and how this relates to the timing of folic acid supplementation. Although in principle this study could be categorised as a case – control design, I would refrain from this expression as there is so much heterogeneity between the countries which were compared in this work. The mortality analyses are of no value to NTD and folic acid supplementation since they do not show the number of deaths due to NTD. Moreover, it was not surprising to see that well developed countries are only seen in Table 4 Stratification by wheat, maize and rice groups is of little interest, given that there is no evidence of a superior efficacy of folic acid from either food. Besides, there are significant variations in the number of countries fortifying folic acid in these foods and their SES. Voluntary fortification with folic acid as well as national recommendations for preconception and first trimester folic acid intake (significant confounders) are not mentioned throughout this manuscript

Some specific comments include:

Keywords: Inclusion of autism is not necessary and confusing Introduction, lines 64-65 – references are missing Introduction, lines 70-71- The sentence is more appropriate for discussion/conclusions Section 2, line 83 – state acronym for parts per million in brackets Section 2, lines 86-87 – state timeframe Line 109 – reference is missing The legends to all tables and figures are poorly described. For example: Table 1 – prevalence is not shown, ppm not explained; Table 2- P value not explained; Figure 6 – excessive legend, no B12 in the pathway although B12 was included in keywords, the term folate refers to natural folate forms, not folic acid. Lines 132-133- poor sentence construction, not sure of the meaning. The lowest folic acid supplementation (ppm) in India seen in Figure 1 and 3 is not discussed The results included in table 3 are not discussed. This table is of highest interest to this work. Discussion, Lines 304-382 – not relevant to this work lines 386-393 Conclusions are not supported by the findings and are not relevant to the title or rationale of this study

Reviewer 2 Report

I would like to commend the authors for using existing data sets to explore a well-known problem in a new creative way. In addition, the text is well written and understandable, and the discussion brings to light many important aspects regarding widespread enrichment of food with folic acid. However, I think the description of the data analysis is somewhat simplified and incomplete. I believe that a better statistical analysis, as well as an explanation of these, would have increased the impact of results and thus provided better support for the conclusion of the article:

Table 1 does not explain how the average in ppm is calculated and what the ± values are. Exactly how was the average calculated? Did the amount folate enrichments vary over years for some countries? What does range mean: the lowest to highest ppm value across countries? All of this should be explained in footnotes and/or text.

Also Tables 2 and 3 lack information on how the NTD average was calculated and what the ± values are. It is also not clear which tests are used to calculate the P-values. It is also unclear what the “Correlation Coefficient vs Not Supplemented" means and how it is calculated. All of this should be explained in footnotes and/or text.

In my opinion, a pooled average with 95% CI using a meta-analysis would be a better way to estimate the average of NTDs and folate enrichment values in Tables 1 and 2. Each country has its background prevalence and its enrichment average and these may also vary over years. This natural between country variations should be taken into account in the overall averages, and this I think a meta-analysis would give you.

In general, I would recommend you use a random effects meta-analysis to better account for variation. Using a meta-analysis, you will also be able to replace Figures 1 to 4 with forest plots. At the same time, you may obtain estimates for heterogeneity between countries, which further provides you some support for pooling the data in the first place.

In sub-figures 1 to 4 (regression analysis) the authors have placed NTD on the x-axis and folate enrichment on the y-axis, which is a bit unusual in my opinion. The convention is to place the response (NTD in this case) on the y-axis, while the explanatory variable (enrichment in this case) on the x-axis. Furthermore, R-squared is not a good measure of association. I suggest the authors replace the correlation coefficient with regression coefficients and confidence intervals, i.e., change in NTD per unit change in folate enrichment. To better account for between-country variation, a “meta-regression” would be an alternative.

In addition to the already performed analyzes, it would be relevant to investigate the association between folate enrichment duration and NTD occurrence. Would that be possible in your material?

It is not always clear when the authors mean supplementation (with pills) or fortification with folate (in food). They use the term "supplementation" for both events. I suggest you distinguish between these two events throughout the manuscript.

Both “prevalence” and “incidence” are used as terms for occurrence. You should choose one of these terms.

Lines 64 and 65 require a reference: "High blood levels of folate can increase ...".

Reviewer 3 Report

The report by Murphy and Westmark is interesting and, despite several limitations partly discussed in the manuscript, indicates that the available evidence is weak that folic acid supplementation at the national level is really effective in preventing neural tube defects in different types of populations.

Major criticisms:

-Abstract. The sentence, "We did not find reduced incidence of NTDs." should be omitted and replaced with the following,"We found a very weak correlation between NTD prevalence and the level of folic acid supplementation irrespective of the cereal grain supplemented compared to the no supplementation population." Moreover, the "Conclusion" of the "Abstract" should be changed accordingly.

-Figure 6, on remethylation and folate cycles is incomplete. Indeed, it does not show vitamin B12, which is only mentioned in the legend. Please, see Figure 3 in a recent Review by Sechi et al., in Nutrition Reviews 2016. To quote this Review may add value to the manuscript.

Minor criticisms:

-Minor English changes are required: e.g., county/country; 5000 ug/day.

Round 2

Reviewer 2 Report

There are some additional points that should be clarified:

In response to the reviewer, the authors describe that the “FFI dataset is a single value for each country regarding 2013 data from the Food and Agriculture Organization of the United Nations”. This information is important for understanding the data analysis and it should be available in the manuscript. What about the country-specific prevalence of NTDs? It is not clear from their response if they have NTD data only for one year or if it is possible to retrieve data for several years per country? Please provide all the information in the methods section or footnotes/titles necessary to understand the strengths and limitations of these data. If the authors have used only one single NTD value (from 2013?) per country in this manuscript, would they expect the main results to be similar for other years before or after 2013? Please discuss.

Figures 1-4 show the prevalence of NTD per 10000 births. I believe that the average data in Table 2 are calculated by summing these country-specific prevalences per 10000 births and then divided by the number of countries within the “fortification-cohort”. However, this is not clear from the text. If the average is calculated in a different way, the country-specific data probably need to be weighted according to population size before averaging. Please confirm or clarify in the methods section exactly how the average data are calculated. Table 2: Confidence intervals indicate statistical significance on the 0.05 level, but in their previous manuscript version, associations were insignificant. So I think there is a misprint somewhere, should there be a minus sign somewhere in the confidence intervals?
